# Waning of anti-SARS-CoV-2 antibodies after the first wave of the COVID-19 pandemic in 2020: A 12-month-evaluation in three population-based European studies

Sophie Novelli[1]*, Christina Reinkemeyer[2,3], Dmitry Bulaev[4], Marc Paul O'Sullivan[4], Chantal J. Snoeck[4], Armin Rauschenberger[4,5], Carmelite Manto[6], Alexey Kolodkin[4], Soumyabrata Ghosh[5], Venkata Satagopam[5], Jerome le Chenadec[6], Karine Barthelemy[7], Stephane Priet[7], Xavier de Lamballerie[7], Andreas Wieser[2,3,8,9], Inge Kroidl[2,8], Michel Vaillant[4], Laurence Meyer[1], Michael Hoelscher[2,3,8,10,11], Noemi Castelletti[2,8,12], Rejko Krüger[4,13,14], Josiane Warszawski[1], for the ORCHESTRA Working Group¶

1 Université Paris-Saclay, UVSQ, Inserm, CESP, APHP Hôpital Bicêtre, Le Kremlin-Bicêtre, France, 2 Institute of Infectious Diseases and Tropical Medicine, LMU University Hospital, LMU Munich, Munich, Germany, 3 Fraunhofer Institute for Translational Medicine and Pharmacology ITMP, Immunology, Infection and Pandemic Research, Munich, Germany, 4 Luxembourg Institute of Health (LIH), Strassen, Luxembourg, 5 Luxembourg Centre for Systems Biomedicine (LCSB), University of Luxembourg, Esch-sur-Alzette, Luxembourg, 6 Université Paris-Saclay, UVSQ, Inserm, CESP, Le Kremlin-Bicêtre, France, 7 Unité des Virus Emergents, UVE, Aix Marseille Université, Inserm, France, 8 German Center for Infection Research (DZIF), Munich, Germany, 9 Max Von Pettenkofer Institute, Faculty of Medicine, LMU Munich, Munich, Germany, 10 Center for International Health (CIH), University Hospital, LMU Munich, Munich, Germany, 11 Unit Global Health, Helmholtz Zentrum München, German Research Centre for Environmental Health (HMGU), Neuherberg, Germany, 12 Institute of Radiation Medicine, Helmholtz Zentrum München, Neuherberg, Germany, 13 Centre Hospitalier de Luxembourg (CHL), Strassen, Luxembourg, 14 Translational Neuroscience, Luxembourg Centre for Systems Biomedicine (LCSB), University of Luxembourg, Esch-sur-Alzette, Luxembourg

¶ Membership of the ORCHESTRA working group is provided in the Supporting Information
* sophie.novelli@inserm.fr

## Abstract

### Objectives

We described waning in anti-SARS-CoV-2 IgG in adult general populations infected during the first wave of the COVID-19 pandemic in 2020 across three European countries.

### Methods

Coordinated analyses were conducted separately in three population-based cohorts with complementary follow-up schedules: the KoCo19 (Germany), EpiCov (France), and CON-VINCE (Luxembourg) cohorts. Serological follow-up was based on the anti-SARS-CoV-2 ELISA-S IgG (Euroimmun) assay. We selected all adults aged 18–79 who had a positive serology (IgG optical density (OD) ratio ≥1.1) between February and July 2020, and at least one subsequent IgG measurement within the following 12 months, while still non-vaccinated.

**Data availability statement:** Koco19: The data that support the findings of this study are available from the KoCo19 collaboration group, but restrictions apply to the availability of these data, which were used under license for the current study, and so are not publicly available. The KoCo19 data are accessible to researchers upon reasonable request to www.koco19.de taking data protection laws and privacy of study participants into account. EpiCov: The second round EpiCov dataset is available for research purpose on CASD (https://www.casd.eu/), after submission to approval of French Ethics and Regulatory Committee procedure (Comité du Secret Statistique, CESREES and CNIL). Access to anonymized individual data underlying the findings may be available before the planned period, on request to the corresponding author, to be submitted to approval of ethics and reglementary Committee for researchers who meet the criteria for access to data. CON-VINCE: The dataset for this manuscript is not publicly available as it is linked to the CON-VINCE Study and its internal regulations. Any requests for accessing the dataset can be directed at con-vince@lih.lu. All data of the manuscript will be provided upon reasonable request and approval by the ethics committee. This study operates under General Data Protection Regulation (GDPR) and Germany, France and Luxembourg data protection laws. The consent forms emphasized implementing appropriate security measures and strictly confidential handling. In addition, data transfers outside the EU require appropriate measures and safeguards. Public release would bypass these transfer control mechanisms. That is why, access to data is possible only by the request procedures that were put in place and approved by the authorities and ethics board of each country, previously described.

**Funding:** This project has received funding from the European Union's Horizon 2020 research and innovation program under grant agreement No 101016167, ORCHESTRA (Connecting European Cohorts to Increase Common and Effective Response to SARS-CoV-2 Pandemic. The views expressed in this paper are the sole responsibility of the authors and the Commission is not responsible for any use that may be made of the information it contains. The KoCo19 study has received funding

## Results

The proportion of seroreversion was 0% within the four first months, based on Koco19 data (n = 65 participants). In the longer term, 31.3% of participants had seroreverted at 6 months (95%CI: 24.4–39.1) (based on EpiCov data, n = 599), 31.3% (95%CI: 11.0–58.7) at 12 months (based on CON-VINCE data, n = 16). From EpiCov data, both baseline low IgG levels and seroneutralization negativity remained predictive of seroreversion in multivariable analysis.

## Conclusion

From population-based cohorts, anti-S IgG levels remained stable during the first 4 months following SARS-CoV-2 infection. Most of the decay occurred afterward; nearly one-third of people seroreverted 6 and 12 months later. Low IgG levels and seroneutralization negativity were independent predictors of seroreversion.

## Introduction

SARS-CoV-2 infection induces a humoral response with the production of IgM, IgA and IgG antibodies [1]. By measuring the level of antibodies in plasma or serum, serological assays allow the identification of individuals infected with SARS-CoV-2 who have produced antibodies in response to the infection.

Serological assays mostly detect IgM/IgG against highly immunogenic viral antigens: the virus surface S (spike) protein that enables interaction (via its RBD domain) and fusion with the target cell, and the nucleocapsid N protein [2]. For both antigens, data accumulated shows that after symptom onset seroconversion of IgM and IgG occurs at about 2–3 weeks and IgM levels drop significantly after 4–6 weeks, whereas IgG titers may last longer, as would be expected after an acute viral infection [3,4]. Antibody levels were found to vary with the severity of symptoms [5,6].

After the onset of the epidemic, the question of what is the long-term duration of humoral immunity against SARS-CoV-2 after natural infection rapidly emerged. Most of the studies that addressed this issue were cohorts of hospitalized patients [7,8] or convalescent individuals enrolled from clinics or hospitals and who mainly experienced moderate to severe COVID-19 disease [3,9–13]. Most recent studies reported that seropositivity for circulating IgG after natural infection in non-vaccinated participants could be maintained for up to 1 year in most people [7,9,14] after onset of COVID-19 symptoms or even longer: 15 months in two studies on patients with mild/moderate to severe COVID-19 disease [10,15], 18 months in a cohort of patients hospitalized with COVID-19 pneumonia in Spring 2020 with a high prevalence of comorbidities [8], 20.5 months after symptoms in a cohort of healthcare workers [16].

Studies on asymptomatic subjects are less numerous and have identified a weaker IgG response and transient persistence of IgG of only a few months in

from the Bavarian State Ministry of Science and the Arts, University Hospital, LMU Munich, Helmholtz Centre Munich, University of Bonn, University of Bielefeld, German Ministry for Education and Research (Project number: 01KI20271 and others). Euroimmun and Roche provided kits and machines for analyses at discounted rates for the study. The EpiCov cohort was supported by research grants from Inserm (Institut National de la Santé et de la Recherche Médicale), the French Ministry for Research, Drees-Direction de la Recherche, des Etudes, de l'Evaluation et des Statistiques, and the French Ministry for Health and by the Région Ile-de-France. The CON-VINCE Study was funded by the National Research Fund Luxembourg (14716281/ CONVINCE/ Kruger) and the André Losch Foundation (Luxembourg). In addition, A.W. and M.H. report personal fees and non-financial support from Roche Diagnostics. A.W. and M.H. report non-financial support from Euroimmun, non-financial support from Viramed, non-financial support from Mikrogen. A.W. and M.H. report grants, non-financial support and other from German Centre for Infection Research DZIF, grants and non-financial support from Government of Bavaria, non-financial support from BMW, non-financial support from Munich Police, nonfinancial support and other from Accenture. A.W. and M.H. report personal fees and nonfinancial support from Dr. Box-Betrobox, non-financial support from Dr. Becker MVZ during the conduct of the study. A.W. is involved in other different patents and companies not in relation to the serology of SARS-CoV-2. A.W. reports personal fees and others from Haeraeus Sensors, nonfinancial support from Bruker Daltonics, all of which are outside the submitted work, and non-srelated to SARS-CoV-2. The funders provided support in the form of salaries for authors, but did not have any additional role in the study design, data collection and analysis, decision to publish, or preparation of the manuscript. The specific roles of these authors are articulated in the 'author contributions' section.

**Competing interests:** AW and MH report personal fees and non-financial support from Roche Diagnostics. AW and MH report non-financial support from Euroimmun, non-financial support from Viramed, non-financial support from Mikrogen. AW and MH report grants, non-financial support and other from German Centre for Infection Research DZIF, grants and non-financial support from

asymptomatic individuals [17,18], with contrasting results explainable by differences in population characteristics and immunoassays [4]. One study reported persistent anti-N IgG seropositivity in 97.6% of blood donors over a median follow-up of 12 weeks, knowing that blood donors are presumably in better health conditions than the overall population [19]. In Iceland, one study in the general population reported no decline in anti-SARS-CoV-2 IgG within 4 months after infection [20]. No data was available with a longer follow-up.

From a public health perspective, serosurveys in population-based cohorts are crucial for estimating the proportion of the population that has been exposed to the virus. They typically consist of repeated sequential antibody measurements in a random sample of individuals, representative of the target population or subpopulation [21–23]. There is little information on the waning of antibody titers in the general population, while this is of major importance for estimating incidence from repeated cross-sectional prevalence surveys.

In the frame of the European ORCHESTRA collaboration, we brought together data from three European population-based cohorts (i.e., the KoCo19 cohort in Munich, Germany, the nationwide EpiCov cohort, France, and the nationwide CON-VINCE cohort, Luxembourg) to estimate the decay of anti-SARS-CoV-2 ELISA-S IgG antibodies against the S1 domain of the viral spike protein among adult individuals infected during the first wave of the COVID-19 pandemic.

## Materials and methods

The three population-based European cohorts launched in April-May 2020 had the initial aim to study the course of the pandemic and monitor the seroprevalence of SARS-CoV-2 in the population. Their respective designs are briefly described below and summarized in Table 1.

For this study, we selected all participants aged over 18 who had a positive IgG ELISA-S (OD ratios > 1.1) result before July 1st, 2020 and had at least one subsequent IgG measurement within the following 12 months. In the three cohorts, collected blood samples were analysed by the same commercial quantitative ELISA kits (Euroimmun®, Lübeck, Germany) for the detection of anti-SARS-CoV-2 antibodies (IgG) against the S1 domain of the viral spike protein (ELISA-S), according to the manufacturer's instructions.

### Cohorts design

**The KoCo19 cohort** [24] is a population-based cohort of 5,313 participants aged ≥14 years randomly chosen from a representative sample of 2,994 households in Munich, Germany between April 5 and June 12 2020. After an initial serological assessment at the baseline visit, participants were followed and tested for SARS-CoV-2 in case of COVID-19-like symptoms. Under the umbrella of the KoCo19 study, individuals with a documented positive SARS-CoV-2 RT-PCR result were then recruited in a prospective longitudinal cohort from April to December 2020 [26,27]. Venous blood was drawn as soon as possible upon infection.

Government of Bavaria, non-financial support from BMW, non-financial support from Munich Police, nonfinancial support and other from Accenture. MH and AW report personal fees and nonfinancial support from Dr. Box-Betrobox, non-financial support from Dr. Becker MVZ during the conduct of the study. AW is involved in other different patents and companies not in relation with the serology of SARS-CoV-2. AW reports personal fees and other from Haereaus Sensors, nonfinancial support from Bruker Daltonics, all of which are outside the submitted work, and non-related to SARS-CoV-2. The funders did not have any role in the study design, data collection and analysis, decision to publish, or preparation of the manuscript. The other authors declare that they have no competing interests.

**Table 1. Summary of the design of the EpiCov, CON-VINCE and KoCo19 cohorts and enrolled participants.**

| Design of each cohort | | Selection of study population and time points for the study | |
|---|---|---|---|
| **Cohort** | **Study design** | **Study population** | **Analysed time points** |
| KoCo19 [24] (Munich, Germany) | **Enrolment**: From April 5 to June 12 2020, enrollment of 5,313 individuals aged ≥14 years <br> **Follow-up**: multiple assessments over 18 months after the first positive RT-PCR test | Individuals who were PCR-tested positive from April to December 2020, had a positive serology test before August 2020 and with at least one second measurement afterwards in November – January 2021 **N = 65** | After natural infection: 1 month (< 30 days), 2 months (30 to < 60 days), 4 months (60 to < 120 days). |
| EpiCov [21] (nation-wide, France) | **Enrolment**: From May 2 to June 1 2020: sampling of 134,391 people aged ≥15 years <br> n = 12,114 respondents with completed serological test (home-self blood sampling) <br> **Follow-up**: 2nd round from October 26 to December 14 2020 | All participants ≥18 years with a positive IgG ELISA-S at the first round (May –June 2020) and who had a second measurement in October - December **N = 599** | 6 months: October – December 2020 |
| CON-VINCE [25] (nation-wide, Lux-embourg) | **Enrolment**: From April 15 to May 5 2020, enrollment of 1,865 individuals >18 years <br> **Follow-up**: 12-month follow-up Blood sampling every 2 weeks for the first 2 months (covering the period May – June 2020) with a final follow-up at 12 months from the initial visit (covering the period April to June 2021) | All participants with positive IgG ELISA-S result in April-June 2020 and who had a second measurement in May-June 2021 without being vaccinated meanwhile **N = 16** | 12 months: Last follow-up time point in April – June 2021 |

The EpiCov study [21] is a national random population-based cohort that combines serological testing from home blood self-sampling and a web/telephone questionnaire. From May 2 to June 1 2020, 134,391 individuals aged ≥15 years living in France were enrolled from a random selection from the FIDELI administrative sampling frame (housing and individual demographic files based on tax files). The FIDELI database is considered to be quasi-exhaustive (>96%) for the population living in France [28]. The EpiCov cohort consists of several rounds. For logistic reasons, only a random subsample (12,114 respondents with serological results) were eligible for home capillary blood self-sampling for serological testing at the first round (May 2nd to June 1st 2020). At the second round (October 26th to December 14th 2020) all respondents of the first round were offered home self-sampling.

The CON-VINCE cohort [25] is a national observational population-based cohort in Luxembourg. It was launched in April 2020 with a particular focus on asymptomatic and oligosymptomatic individuals. From April 15 to May 5 2020, participants (> 18 years) were randomly contacted from a panel of 18,000 people, and from these, 1,865 participants were included upon acceptance through the use of a non-probabilistic web panel. At baseline participants completed a questionnaire and a

laboratory visit which included nasopharyngeal swab sampling for detection of SARS-CoV-2 virus, blood sampling (serum and plasma) for serological testing, and optional stool sampling. Participants were followed from baseline with questionnaires and sampling visits repeated every 2 weeks (4 times) until mid-June 2020. This was followed by a visit with questionnaire only at 11 months, and a visit with questionnaire and sampling at 12 months (annual follow-up).

## Laboratory analyses

In all three cohorts, blood samples were analyzed by the same commercial ELISA kits (Euroimmun®, Lübeck, Germany) for the detection of anti-SARS-CoV-2 antibodies (IgG) against the S1 domain of the viral spike protein (ELISA-S), according to the manufacturer's instructions. The Euroimmun ELISA-S test has a sensitivity of 94.4%, according to the manufacturer's cutoff. It has been evaluated in various studies, which reported a specificity ranging from 96.2 to 100% and sensitivity ranging from 86.4 to 100% [29–32].

According to the threshold specified by the manufacturer, samples with optical density (OD) ratios >1.1 were considered positive (ELISA-S+); samples with OD ratios <0.7 were considered negative.

In addition, in the EpiCov cohort, all samples with an ELISA-S test OD ratio ≥ 0.7 were tested with an in-house microneutralization assay to detect neutralizing anti-SARS-CoV-2 antibodies. For this assay, TMPRSS2-expressing VeroE6 cells cultured in 96-well microplates, 100 TCID50 of the SARS-CoV-2 strain BavPat1 (courtesy of Prof. Drosten, Berlin, Germany) and serial dilutions of serum (1/20–1/160) were used, as described elsewhere [33]. Dilutions associated with the presence or absence of a cytopathic effect on 4.5 days after infection were considered negative and positive, respectively. The virus neutralization titer (VNT) referred to the highest dilution of serum with a positive result. Specimens with a VNT ≥ 40 were considered positive. as the specificity at this threshold was 100% on 486 samples collected before the emergence of SARS- CoV-2 in 2017.

Samples from the 2ⁿᵈ round for which sufficient material remained for additional analyses were analyzed for the detection of antibodies (IgG) against the N-protein of SARS-CoV-2 (Euroimmun®, Lübeck, Germany).

## Ethics

The study was conducted in accordance with the Declaration of Helsinki. The KoCo19 study protocol was approved by the Institutional Review Board of the Medical Faculty at Ludwig Maximilian University Munich, Germany (opinion dated 31 March 2020; number 20–275), prior to study initiation. The EpiCov study was approved by the CNIL (the French data protection authority) (MLD/MFI/AR205138) and the local ethics committee (Comité de Protection des Personnes Sud Mediteranée III 2020-A01191-38). The survey was also reviewed by the "Comité du Label de la Statistique Publique". The CON-VINCE study was approved by the national research ethics committee (Comité National d'Ethique de Recherche, CNER), under reference 202004/01, and by the Luxembourgish Ministry of Health under reference 831x6ce0d. The CON-VINCE study has been submitted for registration on ClinicalTrials.gov (NCT04379297). All participants or their legally authorized representatives had provided informed consent to participation. Written informed consent was obtained from all participants in the CON-VINCE and Koco19 studies. No written consent form was required for EpiCov participants, but all selected participants received a detailed information letter about the questionnaire and serology. In accordance with ethical requirements, consent was considered to have been given if the selected person responded to telephone or to web questionnaire. Completing and sending their blood sample to the laboratory was considered as consent for serology.

## Study design

### Study population

For this study, we selected all participants aged 18 and over who had a positive IgG ELISA-S (OD ratio > 1.1) before July 31st, 2020, and had at least one subsequent IgG measurement during the following 12 months while still unvaccinated.

### IgG repeated measurements

**Time points for follow-up differed in each cohort.** For the KoCo19 cohort, sampling was repeated several times after the first positive RT-PCR in April-June 2020, which allowed us to define three periods for repeated measurements: 1st month (< 30 days), 2 months (30 to < 60 days), 4 months (60 to < 120 days) after first PCR positive test.

For the EpiCov study, the follow-up took place in November 2020, 6 months after enrolment.

For the CON-VINCE cohort, we analyzed follow-up measurements performed at 12 months in April-June 2021 in participants who were not yet vaccinated.

**Outcomes.** In each cohort, we estimated the following quantities for anti-S antibodies at baseline and at the follow-up time points described above: i) the overall IgG level ii) the relative change (%) in IgG level at each follow-up measurement, i.e., the difference in IgG OD ratio from initial time point, divided by IgG ratio at initial time point, iii) the proportion of people who became negative (OD ratio <0.7).

### Statistical analyses

Coordinated statistical analyses were conducted separately in each cohort.

Anti-S IgG levels at each time point and relative change over time were presented as median and interquartile range (IQR). We estimated the proportion of seroreversion, i.e., the proportion of participants with a baseline positive serological test (IgG OD ratio >1.1) in May-July 2020 who became negative (IgG OD ratio <0.7) in each cohort, with its 95% Confidence Interval (CI).

Descriptive analyses were performed overall, and stratified by gender and age. In the EpiCov cohort, estimates were computed taking into account the sampling design, with calibrated weights correcting for non-response, as detailed elsewhere [21]. In the CON-VINCE and KoCo19 cohorts, no weighting procedure was applied.

To investigate whether the persistence of anti-S antibodies could be partially explained by the occurrence of reinfections, anti-N IgG levels were also described 6 months after enrolment in the EpiCov cohort, as a marker of recent infection.

In addition, we studied risk factors of seroreversion, i.e., becoming negative at 6 months, defined as a negative IgG ELISA-S (OD ratio <0.7) at the second round of the EpiCov cohort in November 2020. We performed multivariable logistic regression models, with having a negative ELISA-S test in November 2020 as the dependent variable. Age, gender, IgG level, seroneutralization status (positive versus negative) at baseline (May 2020) and timing of COVID-19-like symptoms onset were included as explanatory variables. COVID-19-like symptoms definition was adapted from the ECDC definition [34] comprising anosmia or dysgeusia, and/or fever with at least cough, dyspnoea or thoracic pain; participants were categorized among three groups: i) asymptomatic, ii) symptomatic before the first lockdown (March 2020), iii) symptomatic between March 2020 and the time of blood sampling on May 2020.

We also studied whether the proportion of seroreversion differed according to persistent post-COVID-19 symptoms, defined as persistence in November 2020 of symptoms (anosmia or dysgeusia, fever, cough, dyspnoea, headache, breathing difficulties, fatigue, muscular pain) which had occurred less than 3 months after the first COVID-19-like symptoms, adapted from the WHO definition [35].

The significance threshold was 0.05. Statistical analyses were performed using R (version 4.0.2) and RStudio (version 2022.02.0) for the CON-VINCE and the KoCo19 cohorts, SAS (version 9.4) and STATA (version 14) for the EpiCov cohort.

## Results

### Study populations

From the KoCo19 cohort, we selected 65 individuals (50.7% female, median age 42 years (IQR, 28 – 56)) who were PCR-tested positive and had a positive ELISA-S serology test between May 2020 and July 31, 2020, and with at least one second measurement afterwards.

From the EpiCov cohort, we selected 599 individuals (62.5% female, median age 42 (IQR, 32 – 53)) who had a positive ELISA-S serology at the first round in May 2020 and a second measurement in November 2020.

From the CON-VINCE cohort, we identified 16 participants (50.0% female, median age 37 (IQR, 32 – 47) who had a positive ELISA-S serology in April-June 2020 and had a second measurement in May-June 2021 without being vaccinated meanwhile.

### Baseline anti-S IgG level and change over time

IgG levels are shown in Table 2, overall and by sex and age group for each cohort. At the baseline measurement, the overall median anti-S IgG level in 18–69 years old participants was 3.7 (IQR, 2.1 – 5.7) in KoCo19, 2.6 (IQR, 1.6 – 4.5) in EpiCov, and 3.6 (IQR, 1.6 – 5.4) in CON-VINCE.

Data from the KoCo19 cohort allowed us to estimate short-term changes in IgG level, with a median relative change of + 12% (IQR, -8 – 35) after 1 month, + 14% (IQR, -11 – 48) at 2 months, and -1% (IQR, -10 – 10) after 4 months (Table 2A).

Longer-term relative changes in IgG level over time were of -55% (IQR, -70 – -30) at 6 months in EpiCov (Table 2B), and -63% (IQR, -78 – -33) in CON-VINCE participants at 12 months (Table 2C).

### Proportion of subsequent SARS-Cov-2 seroreversion

In the KoCo19 cohort, the proportion of participants who reverted to seronegative state was 0% within the 4 subsequent months (with respective upper 95%CI bounds at 1, 2 and 4 months: 8.8%, 13.2%, 19.5%), whereas 31.3% of the EpiCov participants seroreverted by 6 months (95%CI, 24.4 – 39.1) and 31.3% (95%CI, 11.0 – 58.7) of the CON-VINCE participants by 12 months (Table 3). To investigate whether the persistence of anti-S antibodies could be partially explained by occurrence of reinfection, we measured anti-N IgG levels in samples from 523 EpiCov participants in November 2020. Overall, 29% (155/523) had positive anti-N ELISA serology 6 months after enrolment, with a median OD ratio of 1.56 (IQR, 1.29 – 2.17). Among the participants who remained positive for anti-S antibodies, 45% had positive anti-N IgG levels (see S1 Table). Only a limited number of participants (n = 20) showed anti-N OD ratios (>3) and were associated with higher albeit modest IgG anti-S OD ratio (median (IQR) 4.23 (2.48 – 5.19)) compared to the overall population of anti-S positive samples (mean 2.46 (1.15 – 3.82)). Anti-N positivity in November 2020 was associated with higher neutralizing antibody titers in May 2020 (see S2 Table).

### Factors associated with seroreversion

We studied risk factors of seroreversion for anti-S antibodies at 6 months in November 2020 in the EpiCov cohort.

In univariable analysis (Table 4), the probability of seroreversion at 6 months was strongly and inversely associated with baseline IgG level (P < 0.001) and negative seroneutralization assay at baseline. The seroreversion rate at 6 months varied from 61.0% in participants with the lowest IgG levels (1.1 to 1.7) to only 0.1% in the highest quartile of IgG, above 4.69. Among participants who were ELISA-S positive but negative for seroneutralization in May 2020, 54.7% seroreverted in November versus 20.1% of participants who were positive for seroneutralization (P < 0.001). A history of COVID-19-like symptoms was associated with a lower risk of seroreversion at 6 months, especially if symptoms had occurred more recently: 12% of participants who had experienced symptoms between mid-March and May 2020 seroreverted by November, versus 24.7% of those whose symptoms started before mid-March 2020 and 36.9% in those with no symptoms. The risk of seroreversion was also associated with age (P < 0.001): people aged 30–39 became more often seronegative than the other age groups. No relation with gender was observed.

In multivariable analysis, only lower levels of IgG and seroneutralization negativity at baseline in May 2020 remained independently associated with a higher risk of seroreversion in November 2020, after adjustment for age, gender and history of COVID-19 symptoms.

**Table 2. ELISA OD ratios of anti-S SARS-CoV-2 IgG and relative individual changes (%), from first positive serology to subsequent time measurements among adults having positive serology before August 2020 (ELISA IgG OD ratio > 1.1).**

**A) KoCo19 (Munich, Germany)**
**Among participants with IgG OD ratio > 1.1 before August 2020**

| | T0 | | | < 30 days | | | Individual % relative change in IgG OD ratio at < 30 days | |
|---|---|---|---|---|---|---|---|---|
| | N | Median | IQR | N | Median | IQR | Median | IQR |
| ALL | 65 | 3.7 | [2.1 –5.7] | 40 | 4.2 | [2.8 – 5.8] | 12 | [-8 – 35] |
| Gender | | | | | | | | |
| Men | 32 | 3.1 | [2.1 – 5.9] | 17 | 3.9 | [2.6 – 6.2] | 9 | [-8 – 35] |
| Women | 33 | 3.8 | [2.4 – 5.5] | 23 | 4.2 | [3.0 – 5.6] | 15 | [1 – 31] |
| Age (years) | | | | | | | | |
| 18-29 | 29 | 3.8 | [1.7 – 4.4] | 17 | 3.2 | [2.6 – 5.1] | 11 | [-8 – 24] |
| 30-79 | 36 | 3.7 | [2.2 – 6.3] | 23 | 5.5 | [3.8 – 6.7] | 12 | [-6 – 43] |
| | | | | [30 – 60[ days | | | Individual % relative change in IgG OD ratio at [30 – 60 [days | |
| ALL | | | | 26 | 3.4 | [2.8 – 4.8] | 14 | [-11 – 48] |
| Gender | | | | | | | | |
| Men | | | | 14 | 3.5 | [2.9 – 4.8] | 18 | [-1 – 56] |
| Women | | | | 12 | 3.4 | [2.7 – 4.5] | -6 | [-15 – 45] |
| Age (years) | | | | | | | | |
| 18-29 | | | | 12 | 3.3 | [2.4 – 4.6] | 7 | [-13 – 32] |
| 30-79 | | | | 14 | 3.6 | [3.1 – 4.8] | 18 | [-10– 56] |
| | | | | [60 – 120[ days | | | Individual %relative change in IgG OD ratio at [60 – 120[ days | |
| ALL | | | | 17 | 4.9 | [3.3 – 5.6] | -1 | [-10 – 10] |
| Gender | | | | | | | | |
| Men | | | | 10 | 5.1 | [3.5 – 5.6] | -5 | [-19 – 6] |
| Women | | | | 7 | 4.3 | [3.2 – 5.3] | 0 | [-2 – 27] |
| Age (years) | | | | | | | | |
| 18-29 | | | | 5 | 5.2 | [4.2 – 5.7] | -3 | [-8 – -1] |
| 30-79 | | | | 12 | 4.6 | [3.1 – 4.8] | 1 | [-13 – 13] |

**B) EpiCov cohort (France) Among participants with IgG OD ratio > 1.1 before May 2020**

| | | T0 (May 2020) | | 6 months (Nov 2020) | | Individual % relative change in IgG OD ratio at 6 months | |
|---|---|---|---|---|---|---|---|
| | N | Median | IQR | Median | IQR | Median | IQR |
| ALL | 579 | 2.6 | [1.6 – 4.5] | 1.1 | [0.6 – 2.6] | -55 | [-70 – -30] |
| Gender | | | | | | | |
| Men | 217 | 2.6 | [1.5 – 5.0] | 1.0 | [0.6 – 3.0] | -49 | [-65 – -37] |
| Women | 362 | 2.5 | [1.6 – 4.2] | 1.1 | [0.6 – 2.4] | -59 | [-71 – - 26] |
| Age (years) | | | | | | | |
| 18-29 | 72 | 3.1 | [1.7 – 5.0] | 1.3 | [0.7 – 2.5] | -52 | [-63– -23] |
| 30-39 | 122 | 1.9 | [1.5 – 2.7] | 0.6 | [0.4– 1.2] | -66 | [-76 – - 49] |
| 40-49 | 158 | 2.4 | [1.4 – 3.4] | 0.8 | [0.6 – 1.3] | -58 | [- 72 – - 44] |

*(Continued)*

**Table 2.** (Continued)

| A) KoCo19 (Munich, Germany) Among participants with IgG OD ratio > 1.1 before August 2020 | | | | | | | |
|---|---|---|---|---|---|---|---|
| 50-59 | 133 | 4.0 | [2.0 – 7.5] | 2.6 | [0.9 –5.0] | -40 | [-61 – -21] |
| 60-79 | 94 | 3.9 | [1.8 – 6.8] | 2.0 | [0.9 – 3.0] | -49 | [- 73 – - 21] |

| C) CON-VINCE cohort (Luxembourg) Among participants with IgG in OD ratio > 1.1 before August 2020 | | T0 (April - June 2020) | | 12 months (May - June 2021) | | Individual % relative change in OD ratio at 12 months | |
|---|---|---|---|---|---|---|---|
| | N | Median | IQR | Median | IQR | Median | IQR |
| ALL | 16 | 3.6 | [1.6 – 5.4] | 1.4 | [0.7 – 2.1] | -63 | [-78 – -33] |
| Gender | | | | | | | |
| Men | 8 | 3.0 | [1.5 – 5.7] | 0.8 | [0.5 – 1.6] | -78 | [-81 – -41] |
| Women | 8 | 3.7 | [2.4 – 4.6] | 1.5 | [1.3 – 2.1] | -56 | [-69 – -11] |
| Age (years) | | | | | | | |
| 18-29 | 3 | 2.8 | [2.1 – 3.2] | 1.4 | [0.9 – 1.9] | -12 | [-50 – -6] |
| 30-39 | 7 | 3.4 | [1.4 – 4.1] | 1.4 | [0.9 –1.8] | -53 | [-63 – -26] |
| 40-49 | 4 | 3.7 | [2.1 – 6.7] | 0.7 | [0.4 – 1.4] | -80 | [-81 – -78] |
| 50-59 | 2 | 6.7 | [6.4 – 6.9] | 2.9 | [2.1 – 3.6] | -58 | [-68 – -49] |
| 60-79 | 0 | | | | | | |

Of note, participants who reported post-COVID-19 symptoms became less frequently IgG seronegative in November 2020, compared to those who had a symptomatic COVID-19 infection but did not develop post-COVID-19 symptoms or those who had an asymptomatic COVID-19 infection: 8.1% versus 17.0 and 36.9% respectively (P = 0.004).

## Discussion

Here we assessed the waning of anti-SARS-CoV-2 ELISA-S IgG antibodies in adult general populations after the first wave of the COVID-19 pandemic in 2020, using data from three European population-based cohorts with complementary design and follow-up schedules within the European ORCHESTRA collaboration. Based on KoCo19 data, the level of IgG increased within two months following a positive PCR test, although not statistically significant, and then remained stable during the following 60 – 120 days. Based on data from the EpiCov and CON-VINCE cohorts, one-third of participants who tested positive in May 2020 seroreverted 6 and 12 months later, respectively. The median relative decline in IgG titres was -55% and -63% at 6 (EpiCov) and 12 months (CON-VINCE), respectively.

Our results are in line with previous cohort studies of patients that reported a peak in anti-S IgG response at weeks 3 – 4 after infection, followed by a progressive decline after 6 months to stabilize thereafter but remaining detectable [4,15]. Previous studies described the duration of anti-SARS-CoV-2 immunity in healthcare workers, a specific, working age population more exposed to SARS-CoV-2 than the general population. In a UK study, 94% of healthcare workers remained positive over 6 months after a first positive IgG anti-spike test [36]. In a comparable Spanish study, this percentage was of 77% after 9 months [37]. Loesche et al. modelled the kinetics of anti-N antibodies after a first positive PCR result in 2020 in a cohort of employees of the Brigham and Women's Hospital (Boston, MA, USA): the anti-N levels peaked 72 days after infection and the percentage of seroreversion was estimated at 15.1% (7.8 – 25.4) 18 months post-infection [38].

We report here results on waning antibodies in general population after infection during the first wave of the pandemic when the wild-type SARS-CoV-2 was predominant. We therefore cannot draw conclusions on the levels of seroreversion observed after contact with later variants. We applied strong selection criteria by including only participants with positive anti-S IgG OD ratio> 1.1 in May 2020, and defined seroreversion as achieving an OD ratio < 0.7

**Table 3. % of negative anti-S IgG serology (ELISA anti-S IgG OD ratio <0.7) at subsequent time measurements, among adults having positive serology before August 2020 (ELISA anti-S IgG OD ratio > 1.1).**

**A) KoCo19 (Munich, Germany)**

Among participants with ELISA anti-S IgG OD ratio > 1.1 before August 2020

| | % IgG negative at < 30 days | | | % IgG negative at [30 – 60[ days | | | % IgG negative at [60 – 120[days | | |
|---|---|---|---|---|---|---|---|---|---|
| | N | % | Exact 95%CI | N | % | Exact 95%CI | N | % | Exact 95%CI |
| All | 40 | 0.0 | [0 – 8.8] | 26 | 0.0 | [0 – 13.2] | 17 | 0.0 | [0 – 19.5] |
| Gender | | | | | | | | | |
| Men | 17 | 0.0 | [0 – 19.5] | 14 | 0.0 | [0 – 23.2] | 10 | 0.0 | [0 – 30.8] |
| Women | 23 | 0.0 | [0 – 14.8] | 12 | 0.0 | [0 – 26.5] | 7 | 0.0 | [0 – 41.0] |
| Age (years) | | | | | | | | | |
| 18-29 | 17 | 0.0 | [0 – 19.5] | 12 | 0.0 | [0 – 26.5] | 5 | 0.0 | [0 – 52.2] |
| 30-79 | 23 | 0.0 | [0 – 14.8] | 14 | 0.0 | [0 – 23.2] | 12 | 0.0 | [0 – 26.5] |

| **B) EPICOV cohort (France)** | | | | **C) CON-VINCE cohort (Luxembourg)** | | | |
|---|---|---|---|---|---|---|---|
| Among participants with ELISA anti-S IgG OD ratio > 1.1 before May 2020 | | | | Among participants with ELISA anti-S IgG OD ratio > 1.1 before August 2020 | | | |
| | % IgG negative at 6 months (November 2020) | | | % IgG negative at 12 months (May-June 2021) | | | |
| | N | % | (n) | 95%CI | N | % | (n) | 95%CI |
| ALL | 579 | 31.3 | (141) | [24.4 – 39.1] | 16 | 31.3 | (5) | [11.0 – 58.7] |
| Gender | | | | | | | | |
| Men | 217 | 29.1 | (43) | [19.1 – 41.7] | 8 | 50.0 | (4) | [15.7 – 84.3] |
| Women | 362 | 32.4 | (97) | [23.7 – 42.4] | 8 | 12.5 | (1) | [0.3 – 52.7] |
| Age (years) | | | | | | | | |
| 18-29 | 72 | 25.1 | (11) | [8.6 – 54.4] | 3 | 33.3 | (1) | [0.8 – 90.6] |
| 30-39 | 122 | 54.7 | (40) | [39.9 – 68.8] | 7 | 28.6 | (2) | [3.7 – 71.0] |
| 40-49 | 158 | 32.1 | (47) | [21.9 – 44.5] | 4 | 50.0 | (2) | [6.8 – 93.2] |
| 50-59 | 133 | 14.0 | (25) | [7.3 – 25.4] | 2 | 0.0 | (0) | [0.0 – 84.2] |
| 60-79 | 94 | 23.1 | (17) | [11.4 – 41.3] | 0 | | | |

over time. In this way, we aimed to be specific in defining seroreversion. We may have omitted people who were infected at the very start of the pandemic and whose IgG OD ratio was already below 1.1 in May 2020. Our results may therefore underestimate antibody seroreversion rate, and should be interpreted as a minimum estimate of seroreversion in the general population.

Low levels of IgG antibodies and seroneutralizing negativity were found to be independent predictive factors of seroreversion. In univariate analysis, a history of COVID-19-like symptoms, especially if recent, was associated with a lower risk of seroreversion, which can be explained by symptom-associated higher initial levels of IgG serology and seroneutralization activity [5,6]. This is in line with previous reports of lower IgG titres and more rapid waning in asymptomatic individuals compared to symptomatic patients [15,17,37,39,40]. Of note, the highest seroreversion rate (54%) was observed in participants aged 30–39, although this result was no longer statistically significant in the multivariable analysis. This is an unexpected result as young people are usually known to show higher and more persistent antibodies levels than older people. However, in our study, the 30–39 age group already had the lowest IgG levels at enrolment, in May 2020. Since antibody levels are associated with symptom severity (Röltgen et al. 2020; Yan et al. 2022), and younger people had fewer symptoms, their initial antibody production may not have been so high compared with other age groups. Secondly, 30–39-year-olds may have been infected earlier in the pandemic and had already lost some antibodies by the time they were enrolled in EpiCov in May 2020.

**Table 4. Percentage of people with seroreversion in November 2020 (ELISA IgG OD ratio <0.7), among people living in mainland France 2 having a positive serology in May 2020 (ELISA IgG OD ratio > 1.1), according to gender, initial level of anti-S SARS-CoV-2 IgG IgG, seroneutralisation status and symptoms** *(the national EpiCov cohort, rounds 1 & 2).* **People aged between 15 and 17 and over 79 were included in this analysis.**

| | N | (n) | % | 95% CI | OR | 95% CI | P | OR | 95% CI | P |
|---|---|---|---|---|---|---|---|---|---|---|
| | | | Univariable Analysis [3] | | | | | Multivariable Analysis [3,4] | | |
| All with IgG OD ratio > 1.1 in May 2020 | 599 | (141) | 29.5 | [22.8 – 37.2] | | | | | | |
| Initial level of IgG OD ratio | | | | | | | | | | |
| 1st quartile [1.100 -1.700] | 149 | (94) | 61.0 | [47.9 – 72.6] | ref | | <0.001 | ref | | <0.001 |
| 2nd quartile [1.700-2.918] | 150 | (39) | 34.4 | [21.2 – 50.5] | 0.3 | [0.1 – 0.7] | | 0.3 | [0.1 – 0.6] | |
| 3rd quartile }2.918-4.690] | 150 | (7) | 15.1 | [4.1 – 42.3] | 0.1 | [0.02 – 0.5] | | 0.1 | [0 – 0.6] | |
| 4th quartile > 4.690 | 150 | (1) | 0.1 | [0.0 – 0.7] | <0.001 | <0.001 – 0.005] | | <0.001 | <0.001 – 0.001] | |
| Seroneutralization status in May 2020 | | | | | | | | | | |
| Positive (VNT ≥ 40) | 427 | (66) | 20.1 | [25.2 – 39.8] | ref | | <0.001 | ref | | 0.03 |
| Negative (VNT < 40) | 146 | (67) | 54.7 | [14.3 – 27.6] | 4.8 | [2.3 – 9.9] | | 2.8 | [1.1-7.2] | |
| Missing | 26 | | | | | | | | | |
| Gender | | | | | | | | | | |
| Men | 230 | (44) | 25.7 | [16.2 – 38.1] | ref | | 0.36 | ref | | 0.42 |
| Women | 369 | (97) | 31.5 | [23.0 – 41.4] | 1.4 | [0.7 – 2.9] | | 1.4 | [0.6 – 3.2] | |
| Age (years) | | | | | | | | | | |
| 15-29 | 86 | (11) | 19.1 | [6.1 – 46.2] | ref | | 0.001 | ref | | 0.08 |
| 30-39 | 122 | (40) | 54.7 | [39.9 – 68.8] | 5.3 | [1.2 – 22.8] | | 4.7 | [0.9 – 24] | |
| 40-49 | 158 | (47) | 32.1 | [21.9 – 44.5] | 2.1 | [0.5 – 8.7] | | 1.3 | [0.2 – 6.9] | |
| 50-59 | 133 | (25) | 14.0 | [7.3 – 25.4] | 0.7 | [0.2 – 3.3] | | 1.4 | [0.3 – 7.9] | |
| ≥ 60 | 100 | (18) | 22.0 | [11.0 – 39.2] | 1.3 | [0.3 – 6.5] | | 2.4 | [0.4 – 15.7] | |
| Timing of COVID-like symptoms onset | | | | | | | | | | |
| After Mid-March 2020 | 191 | (17) | 12.0 | [4.3 – 29.1] | ref | | 0.03 | ref | | 0.22 |
| Before Mid-March 2020 | 97 | (15) | 24.7 | [10.8 – 46.9] | 2.4 | [0.5 – 10.8] | | 2.2 | [0.5 – 10.3] | |
| No COVID-like symptoms | 311 | (112) | 36.9 | [27.7 – 47.1] | 4.6 | [1.4 – 15.3] | | 2.9 | [0.9 – 9.9] | |

Abbreviations: CI, confidence interval, VNT Virus neutralization testing. 1-Home sampling by finger prick/Euroimmun ELISA-S test. 2- People aged 15 years or over residing in mainland France, outside nursing homes for the elderly and prisons. 3- The sampling design is considered for the estimation of prevalence, confidence intervals (logit transformation) and statistical tests, with the SAS procsurvey procedure. The percentages are weighted by sampling weight (the inverse of inclusion probability), corrected for non-response weights and calibrated on the margin of the census. 4- All variables listed in the column were included in a multiple logistic regression model.

Of interest, in the EpiCov cohort, seroreversion was particularly infrequent (8%) in participants who developed persistent post-COVID-19 symptoms, compared to those who did not. We cannot exclude that in some of these subjects, the association between the persistence of antibodies and post-COVID-19 symptoms could be due to reinfection.

We were not able to assess the incidence of reinfections between IgG measurements. However, several arguments suggest this was uncommon event during the study period. First, the antigenic closeness and protective efficacy confer by the exposure to the variants circulating over the study period, i.e., the D614G variants and then Alpha variant from the end of December 2020 [41–44] makes reinfection unlikely. To go further, we measured in EpiCov participants anti-N antibodies at 6 months after enrolment, which, given their shorter half-life than anti-S antibodies, could be considered markers of recent infections. Six months after enrolment, participants showed low or moderate anti-N IgG levels, even those with positive ELISA-S serology, consistent with a gradual decrease in antibodies over the fairly short period between May and

November 2020, rather than recent infection. Besides, the association of anti-N IgG positivity in November 2020 with sero-protection in May 2020, as shown by neutralizing antibody titers, in May 2020 is in favor of individuals producing a strong antibody response after infection, still detectable 6 months later.

Among the strengths of this study is the collaboration of three European population-based cohorts thanks to the Orchestra collaboration, launched in the same period during the first wave of the COVID-19 pandemic during Spring 2020 and before vaccination was implemented. Our findings could also be generalized to other (Western) European countries during the same period. Moreover, these three cohorts used the same serological assay to detect IgG against the viral spike protein (Euroimmun), while non-standardisation of serological tests used often prevents comparison of results between studies. Most studies investigating waning immunity enrolled participants based on a first positive PCR test, which might result in a selection effect towards people having access to test, likely due to symptoms, in particular at the beginning of the pandemic before the tests became widely available. The cross-sectional design of EpiCov and CON-VINCE cohorts from general populations overcomes this limiting factor while it also allowed us to study antibody waning in participants who experienced no COVID-19-like symptoms (52% of participants with a positive serology in EpiCov and 25% in CON-VINCE).

This study has some limitations. First, we present results from three neighboring countries that experienced COVID-19 waves at the same time, and made close decisions to reduce transmission of the virus. Though the political decisions taken to manage the pandemic were not exactly the same, France being the country most affected and with the most restrictive measures, this contributes to bring together the results from three cohorts. However, each cohort has its own regional or national scope, which explains the differences in sample size and sampling schedules. We took advantage of these differences by showing seroconversion rates separately in each cohort and at different timepoints. Second, we acknowledge that the duration of circulating anti-S IgG levels measured here does not mean duration of protection, though anti-S levels correlate well with levels of neutralizing antibodies [4,15]. Identifying the immune correlates of protection from SARS-CoV-2 infection is an ongoing challenge. Third, for the EpiCov and CON-VINCE cohorts, only one follow-up time point was available, at 6 or 12 months after the initial seropositive test, and we did not have precise information on the timing of infection to evaluate antibody kinetics close to infection, which the Koco19 was able to do. Fourth, the exact timing of infection was unknown for most participants. At the time of the survey, the waning of the SARS-CoV-2 antibodies may have already begun in patients who were infected at the early beginning of the pandemic (before March 2020). This may have resulted in an underestimation of baseline levels and the decrease in anti-S IgG antibodies over time.

## Conclusion

These results from the period before COVID-19 vaccination show that in the general population, antibody titers decline in one-third of individuals from 6 months after natural infection. They contribute to a better understanding of SARS-CoV-2 immunity and provide useful information that need to be accounted for when inferring incidence from data from population-based repeated seroprevalence surveys.

## Supporting Information

**S1 Table. Distribution of ELISA Anti-N OD ratios according to ELISA Anti-S positivity, November 2020** *(the national EpiCov cohort, round 2).*
(DOCX)

**S2 Table. ELISA anti-N IgG OD ratios in November 2020 and Neutralizing antibodies in May 2020 (***the national EpiCov cohort***).**
(DOCX)

**S3 Note. Acknowledgments.**
(DOCX)

## Author contributions

**Conceptualization:** Andreas Wieser, Laurence Meyer, Michael Hoelscher, Rejko Krüger, Josiane Warszawski.

**Formal analysis:** Dmitry Bulaev, Alexey Kolodkin, Jerome le Chenadec, Noemi Castelletti.

**Funding acquisition:** Laurence Meyer, Michael Hoelscher, Rejko Krüger, Josiane Warszawski.

**Investigation:** Chantal J. Snoeck, Armin Rauschenberger, Soumyabrata Ghosh, Venkata Satagopam, Karine Barthelemy, Stephane Priet, Xavier de Lamballerie, Inge Kroidl, Michel vaillant.

**Project administration:** Christina Reinkemeyer, Marc Paul O'Sullivan, Carmelite Manto.

**Writing – original draft:** Sophie Novelli.

**Writing – review & editing:** Sophie Novelli, Christina Reinkemeyer, Dmitry Bulaev, Marc Paul O'Sullivan, Chantal J. Snoeck, Armin Rauschenberger, Carmelite Manto, Jerome le Chenadec, Karine Barthelemy, Xavier de Lamballerie, Andreas Wieser, Michel vaillant, Laurence Meyer, Michael Hoelscher, Noemi Castelletti, Rejko Krüger, Josiane Warszawski.

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
