## [Decision Letter · Decision Letter 0]

28 Oct 2024

PONE-D-24-29510Waning of anti-SARS-CoV-2 antibodies after the first wave of the COVID-19 pandemic in 2020: a 12-month-evaluation in three population-based European studiesPLOS ONE

Dear Dr. Novelli,

Thank you for submitting your manuscript to PLOS ONE. After careful consideration, we feel that it has merit but does not fully meet PLOS ONE’s publication criteria as it currently stands. Therefore, we invite you to submit a revised version of the manuscript that addresses the points raised during the review process.

**Your manuscript has been reviewed and requires modifications prior to making a decision. The comments of the reviewers are included at the bottom of this letter. The Reviewer 1 recommended minor change for your study, and revision on the discussion section. The Reviewer 2 recommended major revision and found methodological problems in the manuscript. I agree with this evaluation and I would, therefore, request for the manuscript to be revised accordingly.**

We look forward to receiving your revised manuscript.

Kind regards,

Asli Suner Karakulah, PhD

Academic Editor

PLOS ONE

**Journal Requirements:**

I have read the journal's policy and the authors of this manuscript have the following competing interests: 

A.W. and M.H. report personal fees and non-financial support from Roche Diagnostics. A.W. and M.H. report non-financial support from Euroimmun, non-financial support from Viramed, non-financial support from Mikrogen. A.W. and M.H. report grants, non-financial support and other from German Centre for Infection Research DZIF, grants and non-financial support from Government of Bavaria, non-financial support from BMW, non-financial support from Munich Police, nonfinancial support and other from Accenture. A.W. and M.H. report personal fees and nonfinancial support from Dr. Box-Betrobox, non-financial support from Dr. Becker MVZ during the conduct of the study. A.W. is involved in other different patents and companies not in relation to the serology of SARS-CoV-2. A.W. reports personal fees and others from Haeraeus Sensors, nonfinancial support from Bruker Daltonics, all of which are outside the submitted work, and non-related to SARS-CoV-2. The other authors declare that they have no competing interests. The funders had no role in the design of the study; in the collection, analyses, or interpretation of data; in the writing of the manuscript; or in the decision to publish the results.

We note that one or more of the authors are employed by a commercial company. 

“The funder provided support in the form of salaries for authors, but did not have any additional role in the study design, data collection and analysis, decision to publish, or preparation of the manuscript. The specific roles of these authors are articulated in the ‘author contributions’ section.”

3. In the online submission form, you indicated that Koco19: The data that support the findings of this study are available from the KoCo19 collaboration group, but restrictions apply to the availability of these data, which were used under license for the current study, and so are not publicly available. Data are however available from the authors upon reasonable request and with permission of the KoCo19 collaboration group.

EpiCov: The second round EpiCov dataset is available for research purpose on CASD (https://www.casd.eu/), after submission to approval of French Ethics and Regulatory Committee procedure (Comité du Secret Statistique, CESREES and CNIL). Access to anonymized individual data underlying the findings may be available before the planned period, on request to the corresponding author, to be submitted to approval of ethics and reglementary Committee for researchers who meet the criteria for access to data.

CON-VINCE: The dataset for this manuscript is not publicly available as it is linked to the CON-VINCE Study and its internal regulations. Any requests for accessing the dataset can be directed at con-vince@lih.lu. All data of the manuscript will be provided upon reasonable request and approval by the ethics committee.

4. One of the noted authors is a group or consortium "the ORCHESTRA working group". In addition to naming the author group, please list the individual authors and affiliations within this group in the acknowledgments section of your manuscript. Please also indicate clearly a lead author for this group along with a contact email address.

Reviewers' comments:

Reviewer's Responses to Questions

**Comments to the Author**

1. Is the manuscript technically sound, and do the data support the conclusions?

Reviewer #1: Partly

Reviewer #2: Yes

2. Has the statistical analysis been performed appropriately and rigorously? 

Reviewer #1: Yes

Reviewer #2: Yes

3. Have the authors made all data underlying the findings in their manuscript fully available?

Reviewer #1: No

Reviewer #2: Yes

4. Is the manuscript presented in an intelligible fashion and written in standard English?

Reviewer #1: Yes

Reviewer #2: Yes

5. Review Comments to the Author

**Reviewer #1:**  The manuscript by Dr Novelli and colleagues aimed to study the evolution of IgG antibodies to SARS-CoV2 Spike antigen over time. To that end, the authors used three cohorts, from Germany, France and Luxemburg, established in the first half of the year 2020 in the three countries. For their study, they included a subpopulation in each of the three cohorts and tested selected plasma samples with an ELISA assay allowing the detection of IgG to SARS-CoV2 S protein.

After testing 65, 599 and 16 samples of the cohorts from Germany, France and Luxemburg, respectively, the authors observed no sero-negativation four months after infection and then a reversion in roughly 1/3 of the patients 6 and 12 months after infection. The authors concluded on the waning of IgG antibodies 6 to 12 months after primary infection in one third of infected persons in their cohorts.

The authors addressed here an important issue in the immunity to SARS-CoV2 and its duration. This type of study might also help address the question of herd immunity. The authors are commended for that.

The work presents also limitations, some of which mentioned by the authors in the discussion. These include:

1. The role of re-infection in the maintenance of IgG antibody level. This point is important and should be discussed further, especially in the light of what is known on covid-19 in the three countries during the study period.

2. Although the 3 countries are neighbors, political decisions during the early phase of covid-19 were not exactly the same and this might have some incidence on epidemiological situation in the three countries. This should be mentioned in the manuscript.

3. The initial cohorts from the three countries included large sample size. However, in the studied population, the sample size per site is very disequilibrated (16, 65 and 599). Did the authors compared these 3 populations? How comparable were they? If they were not comparable, then it is very risky to draw a general conclusion on antibody dynamics from 3 different population in size and structure.

4. Can the authors indicate the sensitivity and specificity of the ELISA assay used?

5. The authors defined two categories in the ELISA results according to OD ratios: below 0.7, the samples were considered negative and above 1.1, they were considered positive. What about those in between? It is written in the manuscript that samples with OD ration above 0.7 were tested for neutralization. How many of those with OD ratio between 0.7 and 1.1 were positive in neutralization (i.e. as per authors’ definition with a titer above 40)? This should be helpful to readers and should be presented.

**Reviewer #2:**  Waning of anti-SARS-CoV-2 antibodies after the first wave of the COVID-19 pandemic in 2020: a 12-month-evaluation in three population-based European studies

In this study, the decrease in antibody levels in unvaccinated Anti-S IgG seropositive individuals from three different centers in Europe was investigated as a retrospective cohort study. The inclusion of Anti-S IgG reactive individuals in the early phase of the COVID-19 Pandemic (in 2020) may be valuable in terms of understanding the effect of natural infection on antibody kinetics in the absence of the vaccine factor. It is appropriate to publish after the necessary major and minor revisions.

Major

1- The principled test used for the virus neutralization test (VNT) should be specified in the Material - Method section. In this context, PRNT (Plaque Reduction Neutralization Test) or Surrogate Neutralization Tests can be used [1].

• In addition, the diagnostic reference values of the neutralization test results should be specified according to the relevant method used (positive and negative limit values).

2- Has a power analysis been performed to support the sample power? If so, it should be stated in the Material-Method section.

3- The variant that was effective at the beginning of the COVID-19 pandemic was D614G. Later, the Alpha (B.1.1.7.) variant was effective. In this context, the development of reinfection in the cases included in the study should be taken into consideration and explained.

4- The reason why Anti-N (Nucleocapsid-specific antibody), which is an important antibody in the post-infection situation, is not preferred instead of Anti-S measurement after SARS-CoV-2 infection should be explained.

5- In the results of the study, it was stated that seroreversion was observed more frequently in the 30-39 age group compared to the 15-29 age group and that it was statistically significant (p<0.001). At this point, it was stated in the discussion section that the 30-39 age group is more socioactive as a possible reason underlying the more frequent seronegativity observed compared to the older age group. This explanation may be possible due to the possibility of early infection. However, in line with basic immunological phenomena, there is more antibody production in young individuals [2, 3]. For example, while 4000 IU/mL of Antibody can be formed in a young individual, 2000 IU/mL of antibody can be formed in an older individual. In this case, it is interesting that a young individual may have early infection but falls into seronegativity earlier or more frequently than an older individual within the elapsed time period. This situation should be clarified from an immunological perspective and further explanations should be provided.

6- As a result, seroreversion was investigated in the cases included in the study in line with decreasing antibodies. Although scientific inferences have been examined very well at this point, recommendations should also be shared in terms of public health. What is recommended for individuals who are seronegative within a certain period of time?

Minor

1- Virus neutralization test principle and commercial kit identification, if used, should be provided.

References

1- Dinç, H. Ö., Demirci, M., Özdemir, Y. E., Sirekbasan, S., Aktaş, A. N., Karaali, R., ... & Kocazeybek, B. (2022). Anti-SARS-CoV-2 IgG and neutralizing antibody levels in patients with past COVID-19 infection: a longitudinal study. Balkan Medical Journal, 39(3), 172.

2- Castellao Tavares, S. M. Q. M., Bravo Junior, W. L., & Alves Leite, J. L. (2015). Analyze the levels of immunoglobulins IgG and IgM in elderly and youngs. IntJ Immunol Immunother, 1 (1).

3- Mattos, M. S., Vandendriessche, S., Waisman, A., & Marques, P. E. (2024). The immunology of B-1 cells: from development to aging. Immunity & Aging, 21(1), 54.

Geri bildirim gönder

Yan paneller

Geçmiş

Kaydedilenler

6. PLOS authors have the option to publish the peer review history of their article (what does this mean? ). If published, this will include your full peer review and any attached files.

**Do you want your identity to be public for this peer review?** For information about this choice, including consent withdrawal, please see our Privacy Policy .

Reviewer #1: No

Reviewer #2: **Yes: ** Bekir Kocazeybek

---

## [Author Response · Author response to Decision Letter 0]

20 Jan 2025

Journal Requirements:

Please ensure that your manuscript meets PLOS ONE's style requirements, including those for file naming. The PLOS ONE style templates can be found at https://journals.plos.org/plosone/s/file?id=wjVg/PLOSOne_formatting_sample_main_body.pdf and https://journals.plos.org/plosone/s/file?id=ba62/PLOSOne_formatting_sample_title_authors_affiliations.pdf

Response: We have formatted the manuscript according to style requirements.

I have read the journal's policy and the authors of this manuscript have the following competing interests:

A.W. and M.H. report personal fees and non-financial support from Roche Diagnostics. A.W. and M.H. report non-financial support from Euroimmun, non-financial support from Viramed, non-financial support from Mikrogen. A.W. and M.H. report grants, non-financial support and other from German Centre for Infection Research DZIF, grants and non-financial support from Government of Bavaria, non-financial support from BMW, non-financial support from Munich Police, nonfinancial support and other from Accenture. A.W. and M.H. report personal fees and nonfinancial support from Dr. Box-Betrobox, non-financial support from Dr. Becker MVZ during the conduct of the study. A.W. is involved in other different patents and companies not in relation to the serology of SARS-CoV-2. A.W. reports personal fees and others from Haeraeus Sensors, nonfinancial support from Bruker Daltonics, all of which are outside the submitted work, and non-related to SARS-CoV-2. The other authors declare that they have no competing interests. The funders had no role in the design of the study; in the collection, analyses, or interpretation of data; in the writing of the manuscript; or in the decision to publish the results.

We note that one or more of the authors are employed by a commercial company.

“The funder provided support in the form of salaries for authors, but did not have any additional role in the study design, data collection and analysis, decision to publish, or preparation of the manuscript. The specific roles of these authors are articulated in the ‘author contributions’ section.”

Response: We have amended the Funding Statement accordingly.

Response: We have included an updated Funding Statement and Competing Interests Statement in the cover letter. Regarding data availability, please see our response below.

3. In the online submission form, you indicated that Koco19: The data that support the findings of this study are available from the KoCo19 collaboration group, but restrictions apply to the availability of these data, which were used under license for the current study, and so are not publicly available. Data are however available from the authors upon reasonable request and with permission of the KoCo19 collaboration group.

EpiCov: The second round EpiCov dataset is available for research purpose on CASD (https://www.casd.eu/), after submission to approval of French Ethics and Regulatory Committee procedure (Comité du Secret Statistique, CESREES and CNIL). Access to anonymized individual data underlying the findings may be available before the planned period, on request to the corresponding author, to be submitted to approval of ethics and reglementary Committee for researchers who meet the criteria for access to data.

CON-VINCE: The dataset for this manuscript is not publicly available as it is linked to the CON-VINCE Study and its internal regulations. Any requests for accessing the dataset can be directed at con-vince@lih.lu. All data of the manuscript will be provided upon reasonable request and approval by the ethics committee.

Response: This study operates under General Data Protection Regulation (GDPR) and Germany, France and Luxembourg data protection laws. The consent forms emphasized implementing appropriate security measures and strictly confidential handling. In addition, data transfers outside the EU require appropriate measures and safeguards.

Public release would bypass these transfer control mechanisms.

That is why, access to data is possible only by the request procedures that were put in place and approved by the authorities and ethics board of each country, previously described.

4. One of the noted authors is a group or consortium "the ORCHESTRA working group". In addition to naming the author group, please list the individual authors and affiliations within this group in the acknowledgments section of your manuscript. Please also indicate clearly a lead author for this group along with a contact email address.

Response: We now have listed the individual authors and affiliations within the ORCHESTRA working group in the acknowledgments section of your manuscript, instead of in an Appendix. We have now indicated Laurence Meyer as the lead author of the ORCHESTRA working group.

Response: We have edited the manuscript accordingly

Reviewer #1:

The manuscript by Dr Novelli and colleagues aimed to study the evolution of IgG antibodies to SARS-CoV2 Spike antigen over time. To that end, the authors used three cohorts, from Germany, France and Luxemburg, established in the first half of the year 2020 in the three countries. For their study, they included a subpopulation in each of the three cohorts and tested selected plasma samples with an ELISA assay allowing the detection of IgG to SARS-CoV2 S protein.

After testing 65, 599 and 16 samples of the cohorts from Germany, France and Luxemburg, respectively, the authors observed no sero-negativation four months after infection and then a reversion in roughly 1/3 of the patients 6 and 12 months after infection. The authors concluded on the waning of IgG antibodies 6 to 12 months after primary infection in one third of infected persons in their cohorts.

The authors addressed here an important issue in the immunity to SARS-CoV2 and its duration. This type of study might also help address the question of herd immunity. The authors are commended for that. The work presents also limitations, some of which mentioned by the authors in the discussion. These include:

1. The role of re-infection in the maintenance of IgG antibody level. This point is important and should be discussed further, especially in the light of what is known on covid-19 in the three countries during the study period.

Response: A limit of our work is that we are not able to assess possible reinfections between IgG measurements. This could be a concern for measurements at 6 and 12 months in the EpiCov and CON-VINCE cohort, respectively. As reinfections induce a rise in antibody production, this could have result in underestimating IgG waning.

However, during the period from July to November 2020, there was no significant antigenic change in the SARS-CoV-2 strains circulating in France (Ding et al. 2022; Wu et al. 2022). The “European” variant D614G probably began circulating in January 2020, and the alpha variant was detected in the United Kingdom in September 2020, and only in December 2020 in France. Furthermore, given the low prevalence in the population at that time, the evolution of the virus was not guided by immunological selection forces, but by forces leading to a gradual increase in transmission. Hence, there is little chance that immunocompetent individuals infected at the start of circulation of the D614G variant in France would have been reinfected within a few weeks by the alpha variant.

Moreover, protective efficacy of previous exposure against viral variants was reported to be very high for variants that circulated at this time of the pandemic (Helfand et al. 2022; Smolenov et al. 2022). A meta-analysis in the setting of the wild-type and the Alpha variant reported that reinfection was an uncommon event (range, 0% to 2.2%).

To complete our results, we also performed additional analyses on samples from EpiCov participants. We measured anti-N IgG levels 6 months after enrolment (November 2020) using Anti-SARS-CoV-2 NCP ELISA kits (Euroimmun®, Lübeck, Germany) for the detection of anti-nucleocapsid SARS-CoV-2 antibodies. As anti-N antibodies have shorter half-life than anti-S antibodies, they are considered in some settings to be marker of recent infection.

Among the 523 participants in whom anti-N IgG levels could have been measured in November-December 2020, 29% (155/523) had positive anti-N ELISA serology. Among these 155 participants, levels of anti-N OD ratio were moderate (median (IQR) 1.56 (1.29 – 2.17)). Only a limited number of participants (n=20) showed “high” anti-N OD ratios (>3) and were associated with higher albeit modest IgG anti-S OD ratio (median (IQR) 4.23 (2.48 – 5.19)) compared to the overall population of anti-S positive samples (mean 2.46 (1.15 – 3.82)). Such distribution was consistent with the presence of residual anti-N antibodies produced after the original infection.

Moreover, anti-N positivity in November 2020 was associated with higher neutralizing antibody titers in May 2020 (see S1 Table). This is in favour of individuals producing a strong antibody response after infection, still detectable 6 months later.

Although, we cannot completely rule out the possibility of few individuals may have been reinfected, the overall picture is rather that of a “classic” gradual decrease in antibodies over the fairly short period between May and November 2020.

We added these elements and discussed these additional results to the manuscript as follows:

Methods/Laboratory analyses (lines 171 – 173):

” Samples from the 2nd round for which sufficient material remained for additional analyses were analyzed for the detection of antibodies (IgG) against the N-protein of SARS-CoV-2 (Euroimmun®, Lübeck, Germany).”

Methods/Statistical analyses (lines 223 – 225):

“To investigate whether the persistence of anti-S antibodies could be partially explained by the occurrence of reinfections, anti-N IgG levels were also described 6 months after enrolment in the EpiCov cohort, as a marker of recent infection”.

Results/ Proportion of subsequent SARS-Cov-2 seroreversion (lines 283 – 291):

“To investigate whether the persistence of anti-S antibodies could be partially explained by occurrence of reinfection, we measured anti-N IgG levels in samples from 523 EpiCov participants in November 2020. Overall, 29% (155/523) had positive anti-N ELISA serology 6 months after enrolment, with a median OD ratio of 1.56 (IQR, 1.29 – 2.17). Among the participants who remained positive for anti-S antibodies, 45% had positive anti-N IgG levels (see S1 Table). Only a limited number of participants (n=20) showed anti-N OD ratios (>3) and were associated with higher albeit modest IgG anti-S OD ratio (median (IQR) 4.23 (2.48 – 5.19)) compared to the overall population of anti-S positive samples (mean 2.46 (1.15 – 3.82)). Anti-N positivity in November 2020 was associated with higher neutralizing antibody titres in May 2020 (see S2 Table).”

Discussion (lines 382 – 393)

“We were not able to assess the incidence of reinfections between IgG measurements. However, several arguments suggest this was uncommon event during the study period. First, the antigenic closeness and protective efficacy confer by the exposure to the variants circulating over the study period i.e., the D614G variants and then Alpha variant from the end of December 2020 (41–44) makes reinfection unlikely. To go further, we measured in EpiCov participants anti-N antibodies at 6 months after enrolment, which, given their shorter half-life than anti-S antibodies, could be considered markers of recent infections. Six months after enrolment, participants showed low or moderate anti-N IgG levels, even those with positive ELISA-S serology, consistent with a gradual decrease in antibodies over the fairly short period between May and November 2020, rather than recent infection. Besides, the association of anti-N IgG positivity in November 2020 with seroprotection in May 2020, as shown by neutralizing antibody titers, in May 2020 is in favour of individuals producing a strong antibody response after infection, still detectable 6 months later.”

References

41. Ding LS, Zhang Y, Wen D, Ma J, Yuan H, Li H, et al. Growth, Antigenicity, and Immunogenicity of SARS-CoV-2 Spike Variants Revealed by a Live rVSV-SARS-CoV-2 Virus. Front Med [Internet]. 7 janv 2022 [cité 15 janv 2025];8. Disponible sur: https://www.frontiersin.org/journals/medicine/articles/10.3389/fmed.2021.793437/full

42. Wu J, Nie J, Zhang L, Song H, An Y, Liang Z, et al. The antigenicity of SARS-CoV-2 Delta variants aggregated 10 high-frequency mutations in RBD has not changed sufficiently to replace the current vaccine strain. Signal Transduct Target Ther. 19 janv 2022;7(1):1‑10.

43. Helfand M, Fiordalisi C, Wiedrick J, Ramsey KL, Armstrong C, Gean E, et al. Risk for Reinfection After SARS-CoV-2: A Living, Rapid Review for American College of Physicians Practice Points on the Role of the Antibody Response in Conferring Immunity Following SARS-CoV-2 Infection. Ann Intern Med. 19 avr 2022;175(4):547‑55.

44. Smolenov I, Han HH, Li P, Baccarini C, Verhoeven C, Rockhold F, et al. Impact of previous exposure to S

---

## [Decision Letter · Decision Letter 1]

16 Feb 2025

Waning of anti-SARS-CoV-2 antibodies after the first wave of the COVID-19 pandemic in 2020: a 12-month-evaluation in three population-based European studies

PONE-D-24-29510R1

Dear Dr. Novelli,

We’re pleased to inform you that your manuscript has been judged scientifically suitable for publication and will be formally accepted for publication once it meets all outstanding technical requirements.

Kind regards,

Asli Suner Karakulah, PhD

Academic Editor

PLOS ONE

Reviewers' comments:

Reviewer's Responses to Questions

**Comments to the Author**

1. If the authors have adequately addressed your comments raised in a previous round of review and you feel that this manuscript is now acceptable for publication, you may indicate that here to bypass the “Comments to the Author” section, enter your conflict of interest statement in the “Confidential to Editor” section, and submit your "Accept" recommendation.

Reviewer #1: (No Response)

Reviewer #2: All comments have been addressed

2. Is the manuscript technically sound, and do the data support the conclusions?

Reviewer #1: Yes

Reviewer #2: Yes

3. Has the statistical analysis been performed appropriately and rigorously? 

Reviewer #1: Yes

Reviewer #2: Yes

4. Have the authors made all data underlying the findings in their manuscript fully available?

Reviewer #1: Yes

Reviewer #2: Yes

5. Is the manuscript presented in an intelligible fashion and written in standard English?

Reviewer #1: Yes

Reviewer #2: Yes

6. Review Comments to the Author

Reviewer #1: (No Response)

Reviewer #2: The authors described waning in anti-SARS-CoV-2 IgG in adult general populations infected during the first wave of the COVID-19 pandemic in 2020 across : From population-based cohorts, anti-S IgG levels remained stable during

the first 4 months following SARS-CoV-2 infection. They concluded that mMost of the decay occurred afterward; nearly one-third of people seroreverted 6 and 12 months later. Low IgG levels and seroneutralization negativity were independent predictors of seroreversion. All my comments are answered and applied to the text.

7. PLOS authors have the option to publish the peer review history of their article (what does this mean? ). If published, this will include your full peer review and any attached files.

**Do you want your identity to be public for this peer review?** For information about this choice, including consent withdrawal, please see our Privacy Policy .

Reviewer #1: No

Reviewer #2: **Yes: ** Bekir Kocazeybek

---

## [Editor Report · Acceptance letter]

PONE-D-24-29510R1

PLOS ONE

Dear Dr. Novelli,

I'm pleased to inform you that your manuscript has been deemed suitable for publication in PLOS ONE. Congratulations! Your manuscript is now being handed over to our production team.

Kind regards,

on behalf of

Dr. Asli Suner Karakulah

Academic Editor

PLOS ONE